# Sediment Bacteria and Phosphorus Fraction Response, Notably to Titanium Dioxide Nanoparticle Exposure

**DOI:** 10.3390/microorganisms10081643

**Published:** 2022-08-13

**Authors:** Sixuan Piao, Donglan He

**Affiliations:** Hubei Provincial Engineering and Technology Research Center for Resources and Utilization of Microbiology, College of Life Science, South-Central Minzu University, Wuhan 430074, China

**Keywords:** titanium dioxide nanoparticles, phosphorus fractionation, diversity decrease, function shift, coexistence pattern

## Abstract

Titanium dioxide nanoparticle (TiO_2_ NP) toxicity to the growth of organisms has been gradually clarified; however, its effects on microorganism-mediated phosphorus turnover are poorly understood. To evaluate the influences of TiO_2_ NPs on phosphorus fractionation and the bacterial community, aquatic microorganisms were exposed to different concentrations of TiO_2_ NPs with different exposure times (i.e., 0, 10, and 30 days). We observed the adhesion of TiO_2_ NPs to the cell surfaces of planktonic microbes by using SEM, EDS, and XRD techniques. The addition of TiO_2_ NPs resulted in a decrease in the total phosphorus of water and an increase in the total phosphorus of sediments. Additionally, elevated TiO_2_ NPs enhanced the sediment activities of reductases (i.e., dehydrogenase [0.19–2.25 μg/d/g] and catalase [1.06–2.92 μmol/d/g]), and significantly decreased the absolute abundances of phosphorus-cycling-related genes (i.e., *gcd* [1.78 × 10^4^–9.55 × 10^5^ copies/g], *phoD* [5.50 × 10^3^–5.49 × 10^7^ copies/g], *pstS* [4.17 × 10^2^–1.58 × 10^6^ copies/g]), and sediment bacterial diversity. TiO_2_ NPs could noticeably affect the bacterial community, showing dramatic divergences in relative abundances (e.g., *Actinobacteria*, *Acidobacteria*, and *Firmicutes*), coexistence patterns, and functional redundancies (e.g., translation and transcription). Our results emphasized that the TiO_2_ NP amount—rather than the exposure time—showed significant effects on phosphorus fractions, enzyme activity, phosphorus-cycling-related gene abundance, and bacterial diversity, whereas the exposure time exhibited a greater influence on the composition and function of the sediment bacterial community than the TiO_2_ NP amount. Our findings clarify the responses of phosphorus fractions and the bacterial community to TiO_2_ NP exposure in the water–sediment ecosystem and highlight potential environmental risks of the migration of untreated TiO_2_ NPs to aquatic ecosystems.

## 1. Introduction

Titanium dioxide nanoparticles are extensively employed in catalysts, composite materials, cosmetics, electronics, food, paints, plastics, sunscreens, and wastewater treatment processes due to their special physicochemical properties [1,2]. TiO_2_ NPs, with explosive demand by society, inevitably migrate into aquatic and terrestrial ecosystems [2,3]. The occurrence of TiO_2_ NPs in the environment has raised public concern about their potential influences on flora and fauna [4,5]. Prior studies have reported that TiO_2_ NPs present distinct toxicities to human cells [6], reduce nematode survival [4], and affect the root split of the carrot [5]. However, the effects of the amount of TiO_2_ NP and exposure time on the composition, diversity, coexistence pattern, and functional redundancy of microorganisms in aquatic ecosystems are poorly understood.

Aquatic microorganisms are responsible for the cycling of key nutrients (e.g., carbon decomposition, nitrogen fixation, and phosphorus mineralization) [7]. Recent studies have reported that TiO_2_ NPs can accelerate methanogenesis in mangrove wetland sediments [8], and affect nitrogen transformation [3] and phosphorus adsorption [9]. However, it remains unknown how TiO_2_ NPs affect phosphorus fractionation in the water–sediment ecosystem.

Excessive phosphorus in aquatic ecosystems is one of the greatest environmental concerns due to its significant contribution to cyanobacterial bloom [10]. The source of phosphorus in an aquatic ecosystem is mainly from external input (e.g., aquaculture wastewater and phosphorus-containing industrial wastewater) and internal input (e.g., phosphorus release from sediments mediated by phosphorus-cycling-related microorganisms) [5,11]. Previous literature studies have reported that engineered TiO_2_ NPs can block phosphorus release from sediments to water by adsorption [9,12,13]. In addition, it has been reported that TiO_2_ NPs show toxicity to planktonic microorganisms by damaging DNA and cells via producing reactive oxygen species [14,15,16], and this might lead to an increase in phosphorus content in the water. Therefore, knowing the environmental risks of TiO_2_ NPs in aquatic ecosystems is important for estimating dynamic changes in phosphorus fractions and is beneficial to appealing to environmental protection policies for mitigating the migration of TiO_2_ NPs to water.

Riding on the boom of high-throughput sequencing, knowledge regarding the composition and diversity of the bacterial community in response to metal oxide nanoparticles is quickly expanding. Most TiO_2_ NP-related studies only report phosphorus adsorption characteristics by TiO_2_ NPs rather than TiO_2_ NP-mediated simultaneous changes in microbial diversity and phosphorus fractions [9,13]. This situation motivated us to estimate the potential environmental risks of TiO_2_ NP migration to aquatic ecosystems. The objectives of this study were to investigate the effects of TiO_2_ NPs on (i) phosphorus fractionation in the sediment–water system, (ii) reductase activity and phosphorus-cycling-related gene abundance, and (iii) composition, diversity, and functions of sediment bacteria. Considering that high-concentration TiO_2_ NPs can induce cell-producing reactive oxygen [1], we hypothesized that bacteria might lose diversity under conditions with high TiO_2_ NP concentrations. To achieve our targets and validate our hypothesis, we employed high-throughput sequencing for the bacterial 16S rRNA gene and determined the content of phosphorus fractions in both water and sediments.

## 2. Materials and Methods

### 2.1. Sample Collection, TiO_2_ NPs Suspension, and Expose Experiment

Sediments were collected from eutrophic Lake Nanhu (Wuhan, China) by using a sludge sampler in July 2019. The visibly large objects (e.g., stones, leaves, and glasses) were removed, and sediments were stirred (to mix them evenly). The sediments contained 3.67% of total carbon, 0.29% of total nitrogen, and 1.21 mg/g of total phosphorus. Lake water was simultaneously collected by using a water sampler (LB-800). The water contained 1.26 mg/L of total phosphorus and 25.33 mg/L of total nitrogen.

Commercially available TiO_2_ NPs (25 nm, purity ≥ 99.8%, anatase) were purchased from Aladdin (Shanghai, China). To prepare 1000 mg/L of TiO_2_ NP stock suspension, 1 g of TiO_2_ NP was added to 1.00 L of Milli-Q water and then the mixture was sonicated at 250 W under 20 °C for 1 h [17].

About 100 g of sediments were added to 250 mL wide-mouthed glass bottles and were spread evenly. Subsequently, 200 mL of lake water was slightly added into the wide-mouthed glass bottles, and we let the mixtures stand for 1 h. According to TiO_2_ NP concentrations used in previous studies [18,19,20], we added different volumes of TiO_2_ NP stock suspensions to these bottles. The final concentrations of TiO_2_ NPs were 5 mg/L (Amount 2 or A2), 10 mg/L (Amount 3 or A3), 20 mg/L (Amount 4 or A4), and 50 mg/L (Amount 5 or A5). The bottles without TiO_2_ NP additions were treated as blank controls (Amount 1 or A1). The bottles in the experimental and control groups were placed in a dark place and incubated at room temperature. These five treatments contained three replicates, and bottles were incubated at room temperature for a total of 30 days. During incubation, we used spoons to collect about 5 g of sediments on days 1, 10, and 30. The collected sediments were stored at −80 °C for subsequent analysis.

### 2.2. Detection of TiO_2_ NPs

To investigate whether TiO_2_ NPs could be adsorbed by aquatic microorganisms, we added 10 mL of 1000 mg/L TiO_2_ NP stock into 10,000 mL of lake water. The lake water without the TiO_2_ NP addition was used as the blank control. The mixture was incubated at room temperature for 24 h; we then collected planktonic microorganisms by filtering the mixture through a 0.22-μm polycarbonate membrane (Millipore Corporation, Billerica, MA, USA). The collected planktonic microorganisms were washed 5 times using sterile water, freeze-dried, and processed further for the detection of TiO_2_ NPs.

Scanning electron microscopy (SEM) and energy dispersive spectroscopy (EDS) were used to characterize cell morphology and the existence of the Ti element by applying a SU8010 Scanning Electron Microscope (HITACHI, Japan). The X-ray diffraction (XRD) patterns of the collected planktonic microorganisms were recorded on a Bruker D8 Advance diffractometer (Bruker, Germany) with graphite monochromatized Cu Ka radiation.

### 2.3. Determination of Phosphorus Fractions and Sediment Enzyme Assay

Water samples in five groups with different TiO_2_ addition amounts were filtered through a 0.22-μm polycarbonate membrane; part of the water was used to measure soluble reactive phosphorus (SRP). The remaining part was digested by potassium persulfate to determine the total soluble phosphorus (TSP) [21]. Water total phosphorus (WTP) was directly digested by potassium persulfate without filtering through a 0.22-μm polycarbonate membrane. Sediment total phosphorus (TP), Olsen P, inorganic phosphorus (IP), organic phosphorus (OP), total apatite inorganic P (AP; P bound to calcium), and non-apatite inorganic P (NAIP; P bound to iron and aluminum oxyhydroxides) were extracted based on standard protocols described in a prior study [22]. Microbial biomass phosphorus (Pmb) was extracted by applying the chloroform fumigation–extraction approach, and a detailed description was reported in a previous study [23]. Phosphorus content was determined by using molybdenum-blue colorimetry. The activities of catalase and dehydrogenase were estimated based on previous studies [24,25].

### 2.4. DNA Extraction, Gene Quantification, MiSeq Sequencing, and Data Processing

The total genomic DNA was extracted from 0.5 g of freeze-dried sediments using the ISO-11063 standardized DNA extraction approach [26]. The extracted DNA was purified, employing a DNA-EZ Reagents M Humic acid-Be-Gone B kit (Sangon Biotech, Shanghai, China) following the manufacturer’s instructions. DNA concentrations were measured applying a NanoDrop 2000 Spectrophotometer (Thermo Fisher Scientific, Waltham, MA, USA). All extracted DNA samples were stored at −80 °C.

The absolute abundances of phosphorus-cycling-related genes in sediment bacteria were determined by employing quantitation polymerase chain reaction (PCR) with the SYBR green mix. Quantitation PCR conditions and primer sequences for amplifying the inorganic phosphorus-solubilizing-related gcd gene, organic phosphorus-mineralizing-related *phoD* gene, and inorganic phosphorus-transporting-related *pstS* gene are summarized in Appendix A.

Universal primers 338F (5′-ACT CCT ACG GGA GGC AGC A-3′) and 806R (5′-GGA CTA CHV GGG TWT CTA AT-3′) were used to amplify bacterial 16S rRNA gene targeting V3-V4 region [27]. The PCR was performed in a thermal cycler (ABI 9700, Thermo, USA) and conducted under the following conditions: a pre-denaturation at 95 °C for 5 min; 30 cycles at 95 °C for 30 s, 58 °C for 40 s, and 72 °C for 40 s; and then a final extension at 72 °C for 10 min. The PCR products were sent for Illumina MiSeq sequencing at the Personal Biotechnology Co., Ltd. (Shanghai, China).

The raw reads were processed using the pipeline of QIIME with the help of Vsearch [28]. To minimize the influences of any random-sequencing errors, we removed (i) sequences that contained ambiguous bases call; (ii) sequences that did not exactly match barcodes and primers; (iii) sequences with average quality scores less than 20; and (iv) sequences with maximum homopolymers less than 10 base pairs. The purified sequences were clustered into operational taxonomic units (OTUs) at 97% similarity levels against the SILVA v132 reference.

### 2.5. Data Analysis

Significant differences, if not specially-stated, were analyzed by analysis of variance, and Tukey’s test was applied to compare the mean values for each variable when data followed a normal distribution (*p* < 0.05). A Venn diagram and nonmetric multidimensional scaling were built to reflect the community composition similarity of bacteria among five groups with different TiO_2_ NP addition amounts by using the “VennDiagram” and “vegan” packages, respectively. Analysis of similarity was applied to estimate whether there were significant differences in the community compositions among the five groups. Permutational multivariate analysis of variance (PERMANOVA) was used to quantitatively evaluate the effects of the TiO_2_ NP amount and exposure time on the phosphorus components, composition, diversity, and function of the bacterial community. Co-occurrence networks were constructed based on Pearson’s correlations between bacterial phyla and the false discovery rate-corrected *p*-values were less than 0.05 [29]. The networks were visualized by using Gephi v. 0.9.2 (https://gephi.org/ (9 June 2022)). Functional prediction of bacterial taxa was performed by applying the ‘‘Tax4Fun2” package in R, and the functional redundancy index (FRI) of each sample was computed according to the 16S rRNA gene sequence similarity [30].

## 3. Results and Discussion

### 3.1. Responses of Phosphorus Fractions to TiO_2_ NP Addition

Phosphorus fractions in both water and sediments varied in different samples with different TiO_2_ NP addition amounts and exposure times (Appendix A), including SRP (0.11–0.27 mg/L), TSP (0.11–0.33 mg/L), WTP (0.33–0.63 mg/L), TP (1.10–1.98 mg/g), Olsen P (0.19–0.62 mg/g), IP (0.84–1.53 mg/g), OP (0.01–0.77 mg/g), AP (0.56–0.55 mg/g), NAIP (0.18–1.05 mg/g), and Pmb (0.19–0.41 mg/g). Most phosphorus fractions presented significant correlations with each other (Appendix A). For instance, SRP was significantly positively correlated with TSP, TP, Olsen P, OP, and AP, and noticeably negatively correlated with WTP and Pmb. According to the results of PERMANOVA, the TiO_2_ NP addition amounts showed significantly greater effects on the phosphorus fractions (R^2^ = 55.60%, F = 56.71; *p* < 0.001) than the exposure times (R^2^ = 3.00%, F = 1.53; *p* > 0.05) (Figure 1). Consequently, the ten phosphorus components presented significant differences among five groups with different TiO_2_ NP addition amounts (Figure 2). The content of SRP, TSP, TP, Olsen P, OP, IP, and AP increased with greater TiO_2_ NP addition amounts, while WTP and Pmb decreased with elevated TiO_2_ NPs. Previous literature has reported that TiO_2_ NPs can enhance phosphorus adsorption by sediments [12]. Moreover, NPs can lead to algae aggregation and precipitation [15,31]. Therefore, the decreased WTP might be via two pathways: (i) direct adsorption by sediments and/or TiO_2_ and (ii) indirect immobilization by TiO_2_ NPs.

Subsequently, we used SEM, EDS, and XRD techniques to investigate the distribution of TiO_2_ NPs. The surfaces of aquatic microorganisms presented smoothly under conditions without the TiO_2_ NP addition and rough under conditions with the TiO_2_ NP addition based on the SEM results (Figure 3A). The Ti element was presented in cell surfaces by the EDS analysis (Figure 3B). The collected microorganism had spinous peaks at 2θ of 25.3°, 36.9°, 39.5°, and 47.9°, which corresponded to anatase (JCPDS 83–2243) (Figure 3C). It was reported that TiO_2_ NPs possess high protein adsorption capabilities [14] and TiO_2_ NPs can bind to algae cells and show rough surfaces [15].

Additionally, it is worth mentioning that the increase in the content of SRP and TSP might be due to the disintegration of aquatic microorganisms. Prior studies have reported that TiO_2_ NPs impose toxicity to cyanobacterium *Synechocystis* sp. by affecting the expression levels of genes (e.g., *psbA*, *psbD*, and *petF*) [16]. The addition of TiO_2_ NPs shows oxidative damage on *Microcystis aeruginosa* by increasing the content of reactive oxygen species and malondialdehyde [15]. In addition, previous literature has also reported that the decomposition of cyanobacteria can contribute to the formation and distribution of iron-bound phosphorus [32]. Therefore, the increases in TP, Olsen P, OP, and AP in sediments might be attributed to phosphorus transfer from water to sediment via algae decomposition and phosphorus precipitation, which can also be proved by significant correlations between sediment phosphorus components and water phosphorus fractions (Appendix A). The increased soluble reactive phosphorus and total soluble phosphorus could be potential phosphorus resources for cyanobacteria growth. Prior studies have reported that TiO_2_ NPs can weaken and even lose their toxicity to organisms by binging other substrates and aging [17,33]. From an environmental risk viewpoint, TiO_2_ NPs might have both positive and negative contributions to cyanobacterial blooms.

### 3.2. Responses of the Bacterial Community to TiO_2_ NP Addition

Microbial biomass phosphorus significantly decreased with a higher TiO_2_ NP addition (Figure 2); we thought that TiO_2_ NPs might have adverse effects on sediment microbial metabolism. According to the results of PERMANOVA, the TiO_2_ NP addition amount (R^2^ = 71.51%, F = 108.93; *p* < 0.001) rather than exposure time (R^2^ = 1.60%, F = 1.22; *p* > 0.05) exhibited significant effects on enzyme activity (Figure 1). Subsequently, we found significant increases in the activities of dehydrogenase (0.19–2.25 μg/d/g) and catalase (1.06–2.92 μmol/d/g) with increases in the TiO_2_ NP addition amounts (*p* < 0.05; Figure 4). Previous literature studies have reported that the presence of metal nanoparticles (e.g., TiO_2_ NPs, Ag NPs, CuO NPs, and NiO NPs) can mediate the generation of reactive oxygen species [15,34], and the oxidative stress shows toxicity to the growth and proliferation of cells [35,36]. Typically, the reactive oxygen species include hydroxyl radical, superoxide, peroxynitrite, hypochlorous acid, nitric oxide, and singlet oxygen [37]. To eliminate reactive oxygen species and survive adverse conditions, microorganisms produce and release reductase (e.g., dehydrogenase, superoxide dismutase, and catalase) to alleviate oxidative stress [38]. The TiO_2_ NP addition amounts (R^2^ = 83.95%, F = 272.47; *p* < 0.001) rather than exposure times (R^2^ = 2.84%, F = 4.60; *p* < 0.05) exhibited significant effects on the enzyme activity (Figure 1). Absolute abundances of phosphorus-cycling-related genes, including *gcd* (1.78 × 10^4^–9.55 × 10^5^ copies/g sediment), *phoD* (5.50 × 10^3^–5.49 × 10^7^ copies/g sediment), and *pstS* (4.17 × 10^2^–1.58 × 10^6^ copies/g sediment), significantly decreased with the increase of the TiO_2_ NP addition amounts (Figure 5). The gene abundances (i.e., *gcd*, *phoD*, and *pstS*) were notably correlated with phosphorus fractions, except NAIP (Table 1). To our knowledge, this is the first study to report that TiO_2_ NPs exhibit diverse effects on phosphorus-cycling-related gene abundance. This phenomenon might be due to the toxicity of TiO_2_ NPs on the sediment bacterial community.

Large differences in the bacterial community composition were found among five groups with different TiO_2_ NP addition amounts (Figure 6). A total of 29,633 OTUs were found in five groups, and they shared 4788 OTUs (Figure 6A). These OTUs were classified into 42 phyla. The *Actinobacteria*, *Proteobacteria*, *Acidobacteria*, and *Chloroflexi* dominated in 45 sediment samples, with relative abundances ranging from 20.70% to 40.29%, from 20.62% to 34.36%, from 7.34% to 20.08%, and from 5.20 to 15.88%, respectively (Figure 6B). Additionally, *Firmicutes*, *Bacteroidetes*, *Gemmatimonadetes*, *Rokubacteria*, *Patescibacteria*, and *Planctomycetes* were secondary dominant phyla. Significant differences were found in relative abundances of *Actinobacteria*, *Acidobacteria*, *Firmicutes*, *Gemmatimonadetes*, *Rokubacteria*, *Patescibacteria*, and *Planctomycetes* among five groups (Appendix A). Moreover, distinct differences in the bacterial community composition were found among five groups based on the nonmetric multidimensional scaling plot, and the analysis of similarity confirmed the differences were significant (R = 0.694, *p* < 0.001) (Figure 6C). According to the PERMANOVA results, exposure time (R^2^ = 20.84%, F = 6.35; *p* < 0.001) exhibited a greater effect on the bacterial community composition compared to the TiO_2_ NP addition amount (R^2^ = 10.54%, F = 6.12; *p* < 0.001) (Figure 1). Sediment total phosphorus showed a larger influence on the bacterial community composition than other phosphorus components based on PERMANOVA (Table 1). Previous literature studies have reported that the TiO_2_ NP addition affected the microbial community composition [1,17,39,40]. Apart from TiO_2_ NPs, other NPs (e.g., ZnO NPs, CuO NPs, and Ag NPs) also showed distinct effects on the microbial community composition [41,42]. This phenomenon might be due to NP toxicity to the activity and living of microorganisms, which in turn affect the microbial community composition.

The taxonomic α-diversity represented by the Shannon–Wiener index (6.5–7.3) showed significant divergences among five groups with different TiO_2_ NP addition amounts (*p* < 0.05; Figure 6D). The TiO_2_ NP addition amount (R^2^ = 69.67%, F = 98.88; *p* < 0.001) exhibited significant effects on the bacterial diversity compared to exposure time (R^2^ = 0.78%, F = 0.57; *p* > 0.05) (Figure 1). The increased TiO_2_ NP addition amount decreased the sediment bacterial diversity, which is inconsistent with prior studies describing that TiO_2_ NPs increased bacterial α-diversity [43,44]. Moreover, it has been reported that the toxicity of TiO_2_ NPs can decrease the diversity and richness of phosphate-accumulating organisms [17]. The divergences in the effects of TiO_2_ NPs on bacterial diversity might be greatly due to the discrepancy in the bacterial community composition in different habitats. Microorganisms show different adaptation capabilities to the environment [45,46]. Furthermore, some microorganisms can produce and release reductase and, thus, might weaken the oxidative stress of TiO_2_ NPs on other microorganisms [15].

We found divergences in coexistence patterns of bacteria in sediments with different TiO_2_ NP addition amounts, showing different topological properties (e.g., node, edge, and average degree) (Figure 7; Appendix A). The core nodes, i.e., those with the highest betweenness centralities, were GAL15 in Amount 1, *Rokubacteria* in Amount 2, WPS-2 in Amount 3, *Verrucomicrobia* in Amount 4, *Bacteroidetes* in Amount 5, and *Armatimonadetes* in all samples. These results imply functions of bacterial communities might vary in five groups. For instance, *Bacteroidetes* phylum harbors the potential to degrade numerous complex carbohydrates [47]. At KEGG pathway level 3, phosphorus-cycling-related functions showed significant differences among five groups with different TiO_2_ NP addition amounts (*p* < 0.05), such as streptomycin-6-phosphatase (EC 3.1.3.39), phosphinothricin acetyltransferase (EC 2.3.1.183), fructose-1,6-bisphosphatase III (EC 3.1.3.11), pyrophosphatase PpaX (EC 3.6.1.1), c-di-GMP phosphodiesterase (EC 3.1.4.52), putative GTP pyrophosphokinase (EC 2.7.6.5), and phosphosulfolactate synthase (EC 4.4.1.19). Consequently, noticeable differences in the community functions at the KEGG pathway level 2 were found among five groups (*p* < 0.05; Figure 8), including biosynthesis of other secondary metabolites, the metabolism of other amino acids, metabolism of terpenoids and polyketides, energy metabolism, lipid metabolism, xenobiotics biodegradation and metabolism, glycan biosynthesis and metabolism, cell motility, membrane transport, translation, transcription, and replication and repair. The exposure time (R^2^ = 36.78%, F = 11.74; *p* < 0.001) showed a significant effect on bacterial diversity compared to the TiO_2_ NP addition amount (R^2^ = 0.62%, F = 0.40; *p* > 0.05) (Figure 1). Prior studies have reported that TiO_2_ NPs can affect bacterial community functions in both aquatic and terrestrial ecosystems [40,48]. More specifically, TiO_2_ NPs can affect nitrogen transformation via adjusting the transcription levels of nitrogen-cycling-related genes (e.g., *nirK*, *nirS*, and *amoA*) [3,17,19] and the abundances of ammonia-oxidizing microorganisms [49]. However, limited studies have reported whether TiO_2_ NPs can affect the transcription levels and abundance of phosphorus-cycling-related genes. In the future, we will explore the effects of TiO_2_ NPs on phosphorus metabolism at both DNA and RNA levels based on pure culture and the community scale.

### 3.3. Potential Effects of Abiotic and Biotic Factors on Phosphorus Fractionation

The addition of TiO_2_ NPs causes phosphorus (i.e., TP, Olsen P, OP, IP, and AP) accumulation in sediments and affects the diversity, composition, and function of the bacterial community. The activity, diversity, and function of the bacterial community might affect phosphorus fractionation in the water–sediment ecosystem. Phosphorus components, except NAIP, were significantly correlated with dehydrogenase and catalase (Table 1). Community diversity was significantly negatively correlated with SRP, TSP, TP, Olsen P, IP, OP, and AP, and dramatically positively correlated with WTP and Pmb (Table 1). The abundances of the top 10 phyla (e.g., *Acidobacteria*, *Firmicutes*, and *Rokubacteria*) showed different correlations with phosphorus components (Appendix A). Additionally, community functions showed significant correlations with phosphorus components in the water–sediment ecosystem, including lipid metabolism, xenobiotics biodegradation and metabolism, and translation and transcription based on KEGG pathway level 2 (Appendix A).

Prior literature studies have reported that microbial activity (e.g., phosphatase and phytase) contributed to organic phosphorus mineralization in a farmland ecosystem [50] and compost system [23]. Bacterial diversity mediates phosphorus immobilization in the South China Sea [51] and can facilitate phosphorus transfer from sediments to water [29]. Some functional bacteria, including phosphate-accumulating bacteria, phosphate-solubilizing bacteria, and phosphate-mineralizing bacteria are responsible for phosphorus transformation in both aquatic and terrestrial ecosystems [11,22,23]. We guess that phosphorus components could affect the diversity, composition, and function of the sediment bacterial community, whereas the bacterial community might affect phosphorus fractionation in the water–sediment ecosystem. Therefore, the effects of TiO_2_ NPs on phosphorus accumulation in sediments might be via two pathways: (i) showing toxicity to planktonic microorganisms and resulting in phosphorus precipitation, and (ii) blocking phosphorus transfer from sediments to water by affecting activity, diversity, composition, and function of the bacterial community in sediments.

## 4. Conclusions

In this study, we characterized the distribution of TiO_2_ NPs and estimated the effects of TiO_2_ NPs on phosphorus fractionation and diversity, composition, and function of the bacterial community in sediments. We found that TiO_2_ NPs could bind to cell surfaces of planktonic microorganisms based on SEM, EDS, and XRD techniques. The addition of TiO_2_ NPs could lead to the accumulation of phosphorus (e.g., total phosphorus and available phosphorus) in sediments, and significantly affect the composition, coexistence pattern, and function of the bacterial community. The absolute abundances of phosphorus-cycling-related genes and bacterial diversity decreased with the increased TiO_2_ NP addition amount, and significantly correlated with phosphorus components. Our findings are of significance for understanding the environmental risks of TiO_2_ NPs. Given the undeniable toxicity of TiO_2_ NPs to organisms, we will conduct more experiments to explore the effects of TiO_2_ NPs on functional microorganisms at both pure culture and community levels in the future.

## Figures and Tables

**Figure 1 microorganisms-10-01643-f001:**
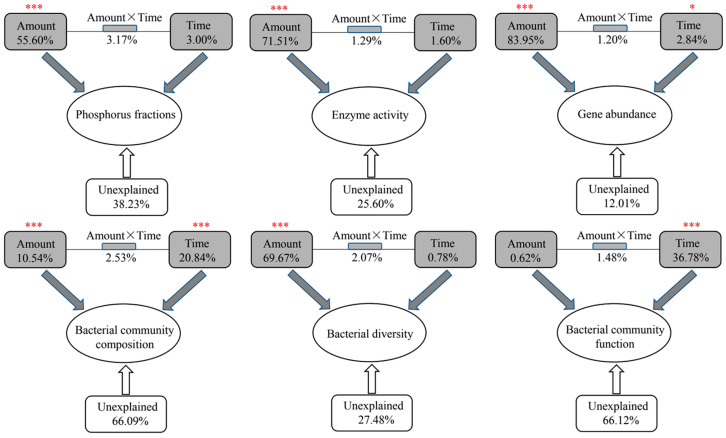
Permutational multivariate analysis of variance (PERMANOVA) showing the effects of TiO_2_ NP addition amounts and exposure times on phosphorus fractions, enzyme activity, phosphorus-cycling-related gene abundance, composition, diversity, and function of the bacterial community in sediments. Asterisks represent significance (*, *p* < 0.05; ***, *p* < 0.001).

**Figure 2 microorganisms-10-01643-f002:**
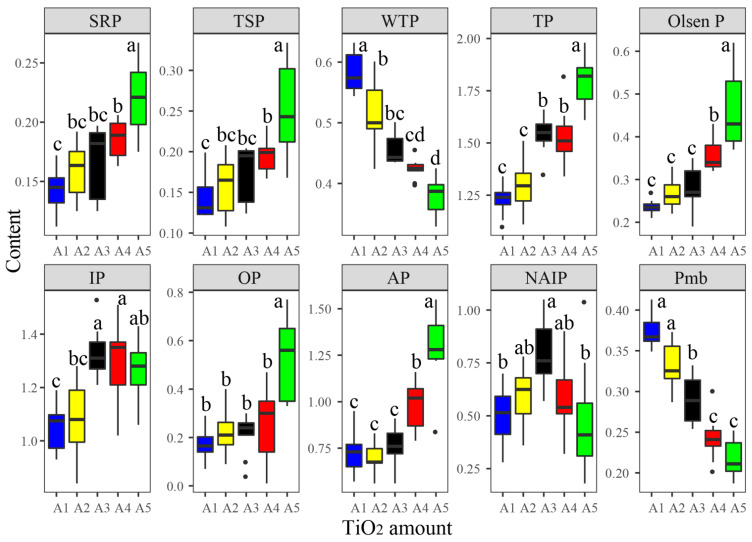
Differences in the content of phosphorus components among five groups with different TiO_2_ NP addition amounts. Different letters above the columns denote significance (*p* < 0.05). Abbreviations: SRP, soluble reactive phosphorus; TSP, total soluble phosphorus; WTP, water total phosphorus; TP, total phosphorus; IP, inorganic phosphorus; AP, total apatite inorganic P; NAIP, non-apatite inorganic P; Pmb, microbial biomass phosphorus.

**Figure 3 microorganisms-10-01643-f003:**
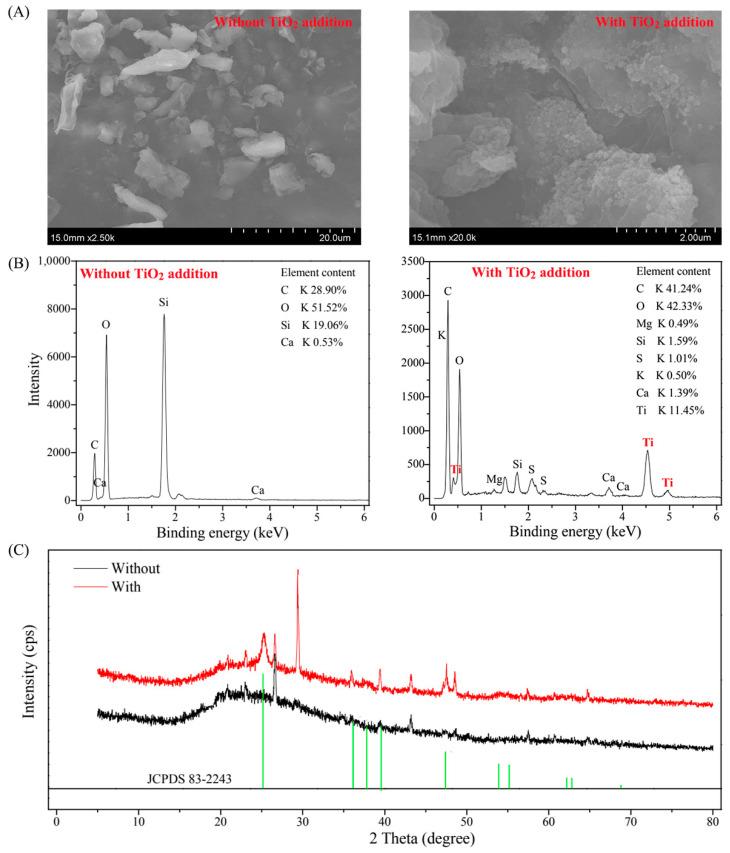
Characterization of the cell surface and TiO_2_ by using scanning electron microscopy (**A**), energy dispersive spectroscopy (**B**), and X-ray diffraction (**C**).

**Figure 4 microorganisms-10-01643-f004:**
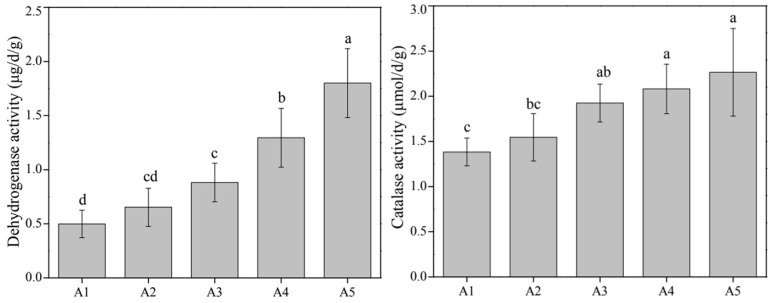
Differences in enzyme activities of dehydrogenase and catalase among five TiO_2_ NP addition amounts (i.e., A1, A2, A3, A4, and A5). Lowercase letters above the columns represent significance (*p* < 0.05).

**Figure 5 microorganisms-10-01643-f005:**
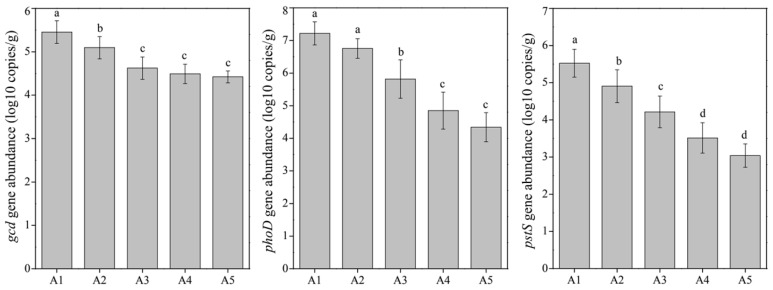
Differences in absolute abundances of phosphorus-cycling-related genes (i.e., *gcd*, *phoD*, and *pstS*) among five TiO_2_ NP addition amounts (i.e., A1, A2, A3, A4, and A5). Lowercase letters above the columns represent significance (*p* < 0.05).

**Figure 6 microorganisms-10-01643-f006:**
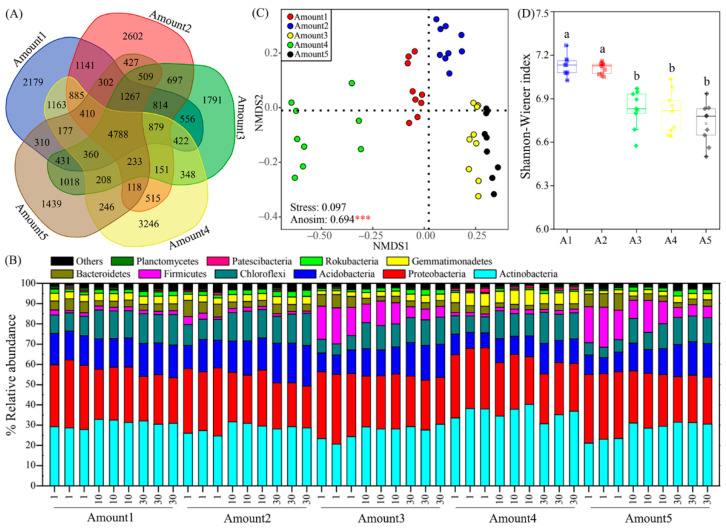
Composition and diversity of the bacterial community in sediments. The Venn diagram shows the shared OTUs of bacteria in five groups with different TiO_2_ NP addition amounts (**A**). Relative abundance of the top ten phyla. Numerical numbers (i.e., 1, 10, and 30) denote the incubating time (**B**). The nonmetric multidimensional scaling plot reflects the bacterial community composition among the five groups. Asterisks denote significance (***, *p* < 0.001) (**C**). Differences in bacterial taxonomic α-diversity (Shannon–Wiener index) among five groups with different TiO_2_ NP addition amounts. Different letters denote differences (*p* < 0.05) (**D**).

**Figure 7 microorganisms-10-01643-f007:**
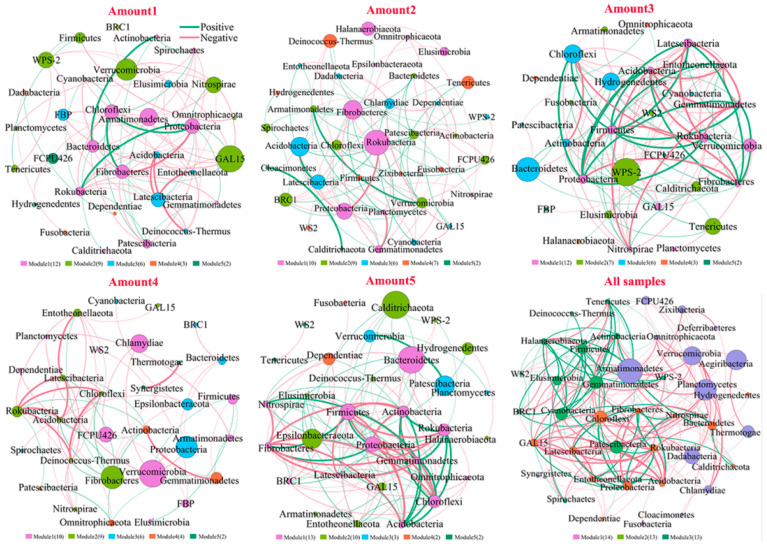
Co-occurrence networks of bacteria in sediments with different TiO_2_ NP addition amounts. Green and red lines represent significantly positive and negative correlations, respectively.

**Figure 8 microorganisms-10-01643-f008:**
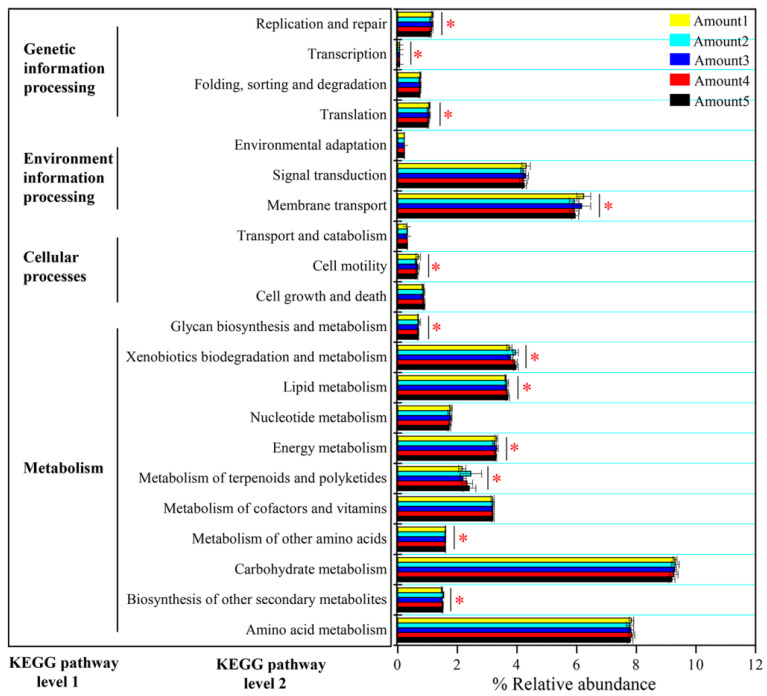
Function profiling showing divergences in bacterial functions at KEGG pathway levels 1 and 2 among five groups with different TiO_2_ NP addition amounts. Asterisks denote significance (*, *p* < 0.05).

**Table 1 microorganisms-10-01643-t001:** Pearson’s correlations between phosphorus fractions and dehydrogenase activity, catalase activity, and taxonomic diversity (Shannon–Wiener index), and the effects of phosphorus fractions on bacterial community composition based on PERMANOVA. Asterisks denote significance (*, *p* < 0.05; **, *p* < 0.01; ***, *p* < 0.001). Abbreviations of phosphorus fractions are defined in the “Materials and Methods” section.

Factor	Dehydrogenase	Catalase	*gcd*	*phoD*	*pstS*	Diversity	Composition
SRP	0.719 ***	0.351 *	−0.679 ***	−0.776 ***	−0.777 ***	−0.391 **	7.52% **
TSP	0.716 ***	0.426 **	−0.640 ***	−0.731 ***	−0.726 ***	−0.346 *	7.21% **
WTP	−0.786 ***	−0.754 ***	0.801 ***	0.792 ***	0.817 ***	0.684 **	8.46% ***
TP	0.778 ***	0.803 ***	−0.743 ***	−0.779 ***	−0.788 ***	−0.763 ***	11.07% ***
Olsen P	0.690 ***	0.580 ***	−0.604 ***	−0.703 ***	−0.709 ***	−0.515 ***	5.75% *
IP	0.476 ***	0.616 ***	−0.661 ***	−0.496 ***	−0.507 ***	−0.545 ***	6.27% *
OP	0.582 ***	0.481 ***	−0.357 *	−0.564 ***	−0.561 ***	−0.495 ***	6.26% *
AP	0.802 ***	0.529 ***	−0.543 ***	−0.543 ***	−0.699 ***	−0.503 ***	5.38% *
NAIP	−0.165	0.213	−0.128	−0.128	0.030	−0.202	2.56%

## Data Availability

The datasets presented in this study can be found in online repositories. The names of the repository/repositories and accession number(s) can be found at https://www.ncbi.nlm.nih.gov/, PRJNA735373.

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
