# Peer review of "Sediment Bacteria and Phosphorus Fraction Response, Notably to Titanium Dioxide Nanoparticle Exposure"

_microorganisms, 2022, doi:10.3390/microorganisms10081643_

Round 1
Reviewer 1 Report
This manuscript is an interesting study evaluating the effect of TiO2 nanoparticles on phosphorus fraction and bacterial community. I think that this paper can be accepted after the following comments have been addressed.
(Line 34) It is recommended that examples of TiO2 used in catalysts or wastewater treatment processes be given as reference. For example, https://doi.org/10.1039/D0EW00787K, https://doi.org/10.1021/acs.est.7b06508.
(Line 171) Although TiO2 nanoparticles are photocatalysts, no information was provided on the light conditions during the experiment. Authors need to provide information about the presence or absence of light and its amount.
(Line 236) Authors need to be clear about what kind of reactive oxygen species might be generated by TiO2 under the given experimental conditions.
Author Response
This manuscript is an interesting study evaluating the effect of TiO2 nanoparticles on phosphorus fraction and bacterial community. I think that this paper can be accepted after the following comments have been addressed.
Response: Thanks for your positive comment.
- (Line 34) It is recommended that examples of TiO2 used in catalysts or wastewater treatment processes be given as reference. For example, https://doi.org/10.1039/D0EW00787K, https://doi.org/10.1021/acs.est.7b06508.
Response: Thanks for your comment.
We have added the required references. The references are as follows:
Lee, C.G.; Javed,H.; Zhang, D.; Kim, J.H.; Westerhoff, P.; Li, Q.; Alvarez, P.J.J. Porous Electrospun Fibers Embedding TiO2 for Adsorption and Photocatalytic Degradation of Water Pollutants. Environ. Sci. Technol. 2018, 52, 4285-4293.
Lee, Y.J.; Lee, C.G.; Kang, J.K.; Park, S.J.; Alvarez, P. Simple preparation method for Styrofoam-TiO2 composites and their photocatalytic application for dye oxidation and Cr(VI) reduction in industrial wastewater.Environ. Sci. Wat. Res. 2021, 7, 222–230.
(2) (Line 171) Although TiO2 nanoparticles are photocatalysts, no information was provided on the light conditions during the experiment. Authors need to provide information about the presence or absence of light and its amount.
Response: Thanks for your comment.
When we collected experimental sediments, we found the water was not very transparent. Therefore, we decided not to use light in the incubation experiments. Please see this sentence “The bottles in experimental and control groups were placed in dark place and incubated in room temperature”. (Line 94–95 in revised manuscript)
(3) (Line 236) Authors need to be clear about what kind of reactive oxygen species might be generated by TiO2 under the given experimental conditions.
Response: Thanks for your comment.
Thanks for your very good suggestion. We considered the potential reactive oxygen during the experiment. However, the determination of reactive oxygen species is not the current research target. We have checked the rest sediments, and found it was not enough for the determination of reactive oxygen species. Besides, we have collected new sediments from lakes, and we are conducting experiments to explore the abundance and composition of reactive oxygen-producing-related microbial community (e.g., psbA, psbD, and petF). Simultaneously, we decide to investigate the reactive oxygen species in this new experiment. The results will be displayed in our ongoing manuscript, please look forward to it.

Reviewer 2 Report
The topic of the article is very forward-looking and relevant, in fact, it would be advisable to examine the ecological effect of not only TiO2 but all nanoparticles, because due to their size, they can play a significant influencing role in living systems.
In an extremely careful way, most of the factors that play a role were examined, so their research extended not only to the quantitative effect of TiO2 but also to the exposure time, and they examined the effect on the amount of phosphorus as well as the genetic effect and even the effect on the diversity of bacteria.
The selected methods are suitable for achieving the set goal. Descriptions are reasonably precise.
The solutions given to explain the results, especially the change in phosphorus types, are correct and are in line with the data reported in previous studies.
The examination of the differences in absolute abundances of phosphorus-cycling-related genes is very exciting and novel, and deserves a more thorough analysis and a closer examination of the relationship with the enzymes.
Contrary to the previous publications, the reason for the difference in the diversity of the bacteria observed cannot be the difference of the microbial fauna, which deviates significantly?
Fig. 1: It very spectacularly summarizes the essential results of the many small experiments carried out, but Some more information would be useful about "Unexplained"
So,it is ready for publication in this form.
Author Response
The topic of the article is very forward-looking and relevant, in fact, it would be advisable to examine the ecological effect of not only TiO2 but all nanoparticles, because due to their size, they can play a significant influencing role in living systems.
In an extremely careful way, most of the factors that play a role were examined, so their research extended not only to the quantitative effect of TiO2 but also to the exposure time, and they examined the effect on the amount of phosphorus as well as the genetic effect and even the effect on the diversity of bacteria.
The selected methods are suitable for achieving the set goal. Descriptions are reasonably precise.
The solutions given to explain the results, especially the change in phosphorus types, are correct and are in line with the data reported in previous studies.
The examination of the differences in absolute abundances of phosphorus-cycling-related genes is very exciting and novel, and deserves a more thorough analysis and a closer examination of the relationship with the enzymes.
Contrary to the previous publications, the reason for the difference in the diversity of the bacteria observed cannot be the difference of the microbial fauna, which deviates significantly?
Fig. 1: It very spectacularly summarizes the essential results of the many small experiments carried out, but Some more information would be useful about "Unexplained"
So,it is ready for publication in this form.
Response: Thanks for your positive comment.